# Recent Advances in Black TiO_2_ Nanomaterials for Solar Energy Conversion

**DOI:** 10.3390/nano13030468

**Published:** 2023-01-24

**Authors:** Lijun Liao, Mingtao Wang, Zhenzi Li, Xuepeng Wang, Wei Zhou

**Affiliations:** Shandong Provincial Key Laboratory of Molecular Engineering, School of Chemistry and Chemical Engineering, Qilu University of Technology (Shandong Academy of Sciences), Jinan 250353, China

**Keywords:** photocatalysis, black TiO_2_, doping, heterojunction, solar energy conversion

## Abstract

Titanium dioxide (TiO_2_) nanomaterials have been widely used in photocatalytic energy conversion and environmental remediation due to their advantages of low cost, chemical stability, and relatively high photo-activity. However, applications of TiO_2_ have been restricted in the ultraviolet range because of the wide band gap. Broadening the light absorption of TiO_2_ nanomaterials is an efficient way to improve the photocatalytic activity. Thus, black TiO_2_ with extended light response range in the visible light and even near infrared light has been extensively exploited as efficient photocatalysts in the last decade. This review represents an attempt to conclude the recent developments in black TiO_2_ nanomaterials synthesized by modified treatment, which presented different structure, morphological features, reduced band gap, and enhanced solar energy harvesting efficiency. Special emphasis has been given to the newly developed synthetic methods, porous black TiO_2_, and the approaches for further improving the photocatalytic activity of black TiO_2_. Various black TiO_2_, doped black TiO_2_, metal-loaded black TiO_2_ and black TiO_2_ heterojunction photocatalysts, and their photocatalytic applications and mechanisms in the field of energy and environment are summarized in this review, to provide useful insights and new ideas in the related field.

## 1. Introduction

With the rapid development of industry and human society, fossil fuels, including coal, natural gas, and petroleum, have been excessively consumed in the last decades, thereby creating energy crises and environmental pollutions. Searching for alternative energy resources and improving the living environment have become urgent issues all over the world. Photocatalytic technology, which can employ the inexhaustible solar energy to H_2_ generation from water splitting [1,2,3], CO_2_ reduction to small sustainable fuels [4,5,6], and environmental pollutant degradation [7,8,9], has been developed and attracted much attention due to a series of excellent physical and chemical characteristics, such as low energy consumption, simple operation, no secondary pollution, low cost, and sustainability [10]. Since 1972, TiO_2_ has been used as a photocatalyst and developed rapidly in the field of energy conversion and environmental remediation [1]. Three main kinds of TiO_2_, including anatase, rutile, and brookite, can be distinguished according to their different crystal structures [11,12,13]. Anatase and rutile are the most frequently investigated TiO_2_ photocatalysts because of their superior photocatalytic activity under UV irradiation than brookite. The photocatalytic performance and properties of TiO_2_ are severely influenced by its preparation, morphology, and dimensions. Serga et al. reported an extraction-pyrolytic method for the synthesis of nanocrystalline TiO_2_ powders using valeric acid as an extractant [14]. This method can be applied for the fabrication of anatase, rutile, or mixed anatase-rutile TiO_2_ powders [14]. Poly (titanium dioxide) is found to have a significant influence on the component compatibility and relaxation behavior of interpenetrating polymer networks [15]. TiO_2_ photocatalysts treated at 800 °C in hydrogen atmosphere for 1 h showed higher visible photocatalytic activity for C-H/C-H coupling of dipyrromethanes with azines than commercial TiO_2_ (P25) [16]. TiO_2_ nanosheets were proved to exhibit superior photocatalytic activity for CO_2_ reduction than the nanoparticle, thanks to its much higher surface area and surface activity [17]. In addition, the effects of the particle size of TiO_2_ on photocatalytic pollutant removal were thoroughly investigated by Kim et al. [18]. The photocatalytic degradation efficiency for methylene blue can be effectively improved by controlling the particle size and TiO_2_ concentration in the reaction mixture [18].

However, due to the wide band gap of TiO_2_ (anatase: 3.2 eV, rutile: 3.0 eV), it can only absorb the ultraviolet part of sunlight (less than 5%), resulting in low light utilization efficiency and low photocatalytic activity [19]. In addition, the high photo-generated electron-hole recombination rate of TiO_2_ materials leads to low quantum efficiency [20]. Therefore, improving the quantum efficiency and photocatalytic activity have always been a concern in the field of photocatalysis.

Previously, the doping modification of TiO_2_, including metal ions (such as Co, Ni, Pt, etc.) or nonmetallic ion (N, H, S, etc.), was introduced to directly modify the TiO_2_ surface electronic properties and broaden the absorption of light, thus improving the efficiency of the charge separation on the TiO_2_ surface [21,22]. Later, TiO_2_ nanomaterials were also combined with other semiconductors to form heterojunctions, thereby enhancing the photo-induced charge separation and migration efficiency, and greatly reducing the corresponding recombination rate [23,24]. In 2011, Chen et al. reported that the hydrogenation strategy can reduce the band gap of TiO_2_, change its color from white to black, expand the light response range to the visible/near infrared region, and improve the visible light catalytic performance [25]. This partially reduced TiO_2_ is coined as “black TiO_2_” in the study [25]. Since then, studies on black TiO_2_ in various fields of photocatalysis, including energy conversion and pollutant removal, have been growing over the last decade. The above-mentioned modification methods (such as doping, heterojunction, etc.) were subsequently applied to black TiO_2_ to narrow the band gap, thereby further promoting the visible light absorption and charge separation efficiency. 

In this review, the structure properties, synthesis routes, and applications of black TiO_2_-based nanomaterials in the environmental and energy fields, such as photocatalytic water splitting and the photodegradation of organic pollutants, are summarized. As shown in Figure 1, the above aspects will be concluded and discussed from the perspectives of black TiO_2_, doped black TiO_2_, metal-loaded black TiO_2_, and black TiO_2_ heterojunction. Finally, the status quo of black TiO_2_ materials is reviewed, and the future development prospects and challenges are proposed.

## 2. Morphology and Structural Properties of Black TiO_2_

The beauty of nanomaterials is that the (photo)catalytic activity is highly influenced by their morphology and structural properties, including the crystal structure, presence of vacancies, and partially phase transformation. Scanning electron microscopy (SEM) and transmission electron microscopy (TEM) were frequently utilized to investigate the morphology of black TiO_2_. Black TiO_2_ with different morphologies, such as nanospheres [26,27,28,29,30,31,32,33,34], nanotubes [35,36,37,38], nanoarrays [39,40,41,42,43], nanowires [44,45,46,47], nanoplates [48,49], nanosheets [50,51], nanobelts [52,53], nanocages [54], nanoflowers [55], nanofibers [56], hollow shells [57,58,59], films [60,61], and nanolaces [62], were synthesized via varied synthetic approaches. Commercial TiO_2_ materials are often directly used to produce black TiO_2_ nanospheres. Biswas et al. obtained black TiO_2-x_ nanoparticles at high temperatures via NaBH_4_ reduction and studied their light absorption ability after reduction [28]. The band gap of black TiO_2_ nanospheres was 2.54 eV, which was much lower than the pristine commercial anatase (3.27 eV) [28]. Katal et al. prepared black TiO_2_ nanoparticles at high temperatures under a vacuum atmosphere and investigated the color change and shrinkage of reduced TiO_2_ pellets over temperature [63]. The reduction process was performed by sintering commercial P25 pellets under a vacuum condition at different temperatures (500, 600, 700, 800 °C) for 3 h [63]. The color of the white P25 pellets changed into pale yellow after calcination at 500 °C in the air condition [63]. Its color became darker after sintering at 500 °C under the vacuum condition [63]. Black TiO_2_ pellets were obtained after calcination at temperatures higher than 500 °C, and the size of the pellets became smaller after treatment at 700 °C [63]. The phase transformation from anatase to rutile was observed in TiO_2_ after high temperature calcination [63]. The corresponding visible change and red-shift in UV-vis absorption spectra are presented in Figure 2A [63]. The band gap energy gradually decreased with temperature from 3.1 eV for P25, to 2.24 eV for BT-800 [63].

The synthesis of black TiO_2_ with unique morphology, such as nanoflowers, tubes, and wires, usually necessitates specific synthetic procedure for TiO_2_ nanomaterials, including hydrothermal, solvothermal treatment, anodization, etc. Lim et al. prepared partially reduced hollow TiO_2_ nanowires (R-HTNWs) using the hydrothermal method and the subsequent treatment with NaBH_4_ under the nitrogen atmosphere [47]. The local distribution of Ti^3+^ species (oxygen vacancies) in reduced hollow TiO_2_ nanowires was confirmed to be primarily present in the surface region compared to the core using electron energy loss spectroscopy (EELS) [47]. In addition, trace impurities including B, Na, N from NaBH_4_, and nitrogen were located mostly at the surface and the distorted rutile structure region of R-HTNWs [47]. The SEM, TEM image, and EELS Ti L_2,3_ data are illustrated in Figure 2B [47]. Ti^3+^ present on the surface of TiO_2_ could be stabilized by the surface impurities [47]. Black TiO_2_ materials generally possessed certain amounts of oxygen vacancies, which can be confirmed by X-ray photoelectron spectroscopy (XPS). The concentration of oxygen vacancies was normally controlled by the different thermal treatment time or temperature [41,45,51]. However, there is lack of precise, quantitative characterization techniques for oxygen vacancies present on the black TiO_2_ surface. The band gap parameters of black TiO_2_ were usually measured and calculated by XPS and UV-vis spectroscopy measurements. The decrease in Eg of black TiO_2_ was assumed to be related to the surface disorder, including the presence of Ti^3+^ and oxygen vacancies [47].

The mesoporous structure of TiO_2_ can increase the surface area and phase stability. Zhou et al. synthesized mesoporous black TiO_2_ hollow spheres (MBTHSs) via the combination of a template-free solvothermal method and amine molecules encircling strategy, and the subsequent atmospheric hydrogenation process [64]. The wall thickness and diameter of MBTHSs could be tuned by adjusting the solvothermal reaction time and the Ti precursor concentration, respectively [64]. Ti^3+^ species were proved to be mainly present in the bulk but not on the surface of MBTHSs via XPS measurements [64]. The light absorption of MBTHSs was effectively extended to the visible light range compared with the pristine TiO_2_ [64]. The synthetic procedure, SEM image, and UV-vis absorption properties are present in Figure 2C [64]. The anatase phase remained unchanged after hydrogenation [64]. The band gap of mesoporous TiO_2_ was largely reduced to 2.59 eV after hydrogenation [64]. The black TiO_2_ consisted of mesoporous structure with cylindrical channels providing the relatively high surface area of ~124 m^2^ g^–1^ [64]. They also fabricated the heterojunctions of γ-Fe_2_O_3_ nanosheets/mesoporous black TiO_2_ hollow sphere to enhance the charge separation and photocatalytic tetracycline degradation efficiency [59]. Porous black TiO_2_ photocatalysts tended to appear in three dimensional structures, such as foams, pillars, and hollow structures. Zhang et al. synthesized the 3D macro-mesoporous black TiO_2_ foams via freeze-drying, cast molding technology, and high-temperature surface hydrogenation [65]. The large, closed pores were generated using polyacrylamide as the organic template, while plenty of open pores were formed in the frameworks and on the surface of the black TiO_2_ thanks to the water evaporation in the freeze-drying process [65]. This black TiO_2_ material exhibited a self-floating amphiphilic property and an enhanced solar energy harvesting efficiency [65]. Zhou et al. prepared porous black TiO_2_ pillars through an oil bath reaction and high-temperature hydrogenation reduction [66]. The porous structure and mesopores of black TiO_2_ pillars were clearly observed by the Scanning electron microscope and transmission electron microscope [66]. The enhanced photocatalytic performance was attributed to more active surface sites offered by the porous pillar structure and the self-doped Ti^3+^ [66]. The hollow structured black TiO_2_ with plenty of pore channels and an exposed surface also showed an enhanced photocatalytic efficiency [57]. The pores of the porous black TiO_2,_ generally located in its whole frameworks with open pores connected with surface, providing abundant active sites and surface defects, thus promoting the photocatalytic performance. Ethylenediamine was often utilized to maintain the porous structure of black TiO_2_ and to prevent its phase transformation from anatase to rutile.

In addition to oxygen vacancies, disordered structures and surface amorphization in black TiO_2_ may have significant impacts on its photoresponsive properties. Kang et al. prepared black TiO_2_ with amorphous domains through a glycol-assisted solvothermal method and subsequent calcination [67]. Oxygen vacancies were introduced in the amorphous domains of the black TiO_2_ nanosheets [67]. Figure 2D shows the color and optical absorption property changes [67]. The color of the brown TiO_2_ turned into black after 2 h of calcination at 350 °C under Ar [67]. The light absorption of black TiO_2_ was significantly extended to the near-infrared region [67]. Oxygen vacancies were confirmed to be present in the subsurface of black TiO_2_ by first-principle calculations [67]. Table 1 summarizes the properties of some black TiO_2_ nanomaterials with varied morphology. The color of most reduced TiO_2_ is black. The morphology of black TiO_2_ is not determined by the reduction process.

**Figure 2 nanomaterials-13-00468-f002:**
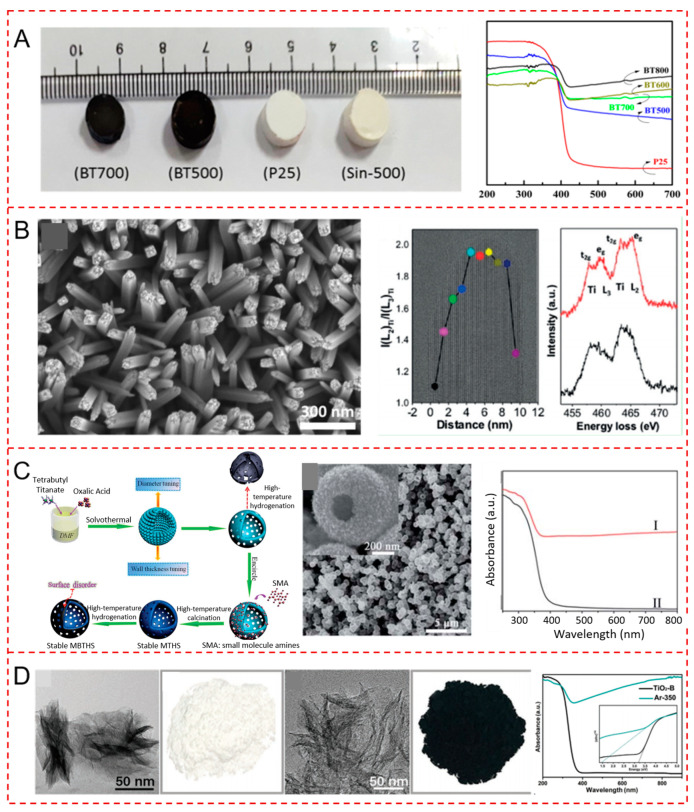
(**A**) Picture and UV-vis spectra of reduced P25 pellets at various temperatures. Reprinted with permission for ref. [63]. Copyright 2018, American Chemical Society. (**B**) SEM and TEM image and overlaid I(L_2_)_Ti_/I(L_3_)_Ti_ values determined from the EELS Ti L_2,3_ data at each position of R-HTNWs. Reprinted with permission for ref. [47]. Copyright 2019, Wiley-VCH. (**C**) Schematic illustration of the synthetic procedure, SEM image, for MBTHSs and UV-vis absorption properties (I: mesoporous black TiO_2_ hollow spheres; II: mesoporous TiO_2_ hollow spheres). Reprinted with permission for ref. [64]. Copyright 2016, The Royal Society of Chemistry. (**D**) TEM images, and optical photographs of (black) TiO_2_, and UV-vis spectra. Reprinted with permission for ref. [67] Copyright 2021, Wiley-VCH.

## 3. Synthesis of Black TiO_2_

Currently, various methods were used to synthesize black TiO_2_, which can be divided into two main approaches: high temperature hydrogenation reduction and solid phase reduction [68]. The high temperature hydrogenation method often uses hydrogen or hydrogen-contained gas mixtures to treat samples at high temperatures [69,70]. The materials used in the solid phase reduction method are generally NaBH_4_ [71], CaH_2_ [72], Mg powder [72], or other reducibility materials [72]. The reduction method can be expressed in reaction Equation (1):TiO_2_ + A → TiO_2−x_ + AO_x_(1)

In addition, researchers also use hot wire annealing [73], laser irradiation [74,75], anode reduction [76], and other methods to synthesize black TiO_2_ [77,78].

### 3.1. High Temperature Hydrogenation

Hydrogen reduction involves the reduction of pure H_2_ gas, H_2_/Ar, or H_2_/N_2_ mixture at high or low pressures [69], which is a simple, effective, and straightforward method. 

Zhou et al. successfully prepared the ordered mesoporous black TiO_2_ material by hydrogenation at high temperature (500 °C) under atmospheric pressure (Figure 3), which had a larger specific surface area and pore size compared with the pristine titanium dioxide [79]. As shown in Figure 3, after hydrogenation at high temperature, the regular hexagonal channel of the obtained black TiO_2_ was completely maintained [79]. It can be seen from the XRD in Figure 3 that there was no phase change in black TiO_2_ compared with the original materials, thus proving the high thermal stability of the sample prepared by this method [79]. Notably, it can also be clearly seen that its crystallinity decreased, proved by the XRD intensity, thereby indicating that the surface disorder of TiO_2_ has been created after the hydrogenation process [79]. The color of the white TiO_2_ turned into black after 3 h of hydrogenation [79].

Black TiO_2_ with different morphologies can be obtained via hydrogenation. Yang et al. prepared one-dimensional black TiO_2_ nanotubes by the hydrogenation method, with an inner diameter of 7 nm and a wall thickness of 6 nm, as presented in Figure 4A [80]. Spherical and lamellar structures have also received much attention due to their large specific surface areas. As shown in Figure 4B, after simple hydrogenation reduction, Li et al. successfully prepared black TiO_2_ nanospheres and observed the mesoporous structure in the TEM image [81]. Although the crystal surface structure of anatase became slightly disordered after hydrogenation, its special lattice fringes (d = 0.35 nm) did not change [81]. Black TiO_2_ nanotubes with the mesoporous nanosheet structure were successfully prepared by the hydrogen reduction method by Zhang et al. [82]. Ethylenediamine coating method was used before hydrogenation [82]. The original morphology of TiO_2_ was completely retained [82]. Wu et al. also synthesized two dimensional ultrathin mesoporous black TiO_2_ nanosheet materials using the similar ethylenediamine encircling strategy (Figure 5) with 4 h hydrogenation reaction at 500 °C [83].

In addition to the method of hydrogen reduction at high temperature and atmospheric pressure, researchers also use the high pressure method. Wu et al. prepared black anatase TiO_2_ in a two-step process [84]. The sample was degassed at 200 °C firstly, then was heated to 400 °C, and hydrogenated under high pressure (5-bar) for 24 h. The black sample was finally obtained after cooling to room temperature. The black TiO_2_ was successfully prepared by Hamad et al. at a high pressure (8 bar) and relatively mild temperature [85]. The hydrogenation time was longer than in other similar research (1–5 days) [85]. The synthesized samples were uniform and stable in size, and showed higher photocatalytic activity compared with the pristine white TiO_2_ [85]. Mixed gases with H_2_ were also used as reducing agents in the synthesis of black TiO_2_. Cai et al. successfully prepared black TiO_2_ with the surface disorder structure using H_2_/N_2_ mixed gas with 10% content of hydrogen [86].

### 3.2. Solid Phase Reduction

Compared with the high temperature hydrogenation, the solid phase reduction method has certain advantages. The high temperature hydrogenation process normally starts from the outside to the inside with a relatively moderate reaction rate, while the solid phase reduction method can provide a more complete and intense reaction and may produce a series of doping at the same time. The defect is a double-edged sword. Too many defects may be detrimental to the photocatalytic performance, so the proportion and dosage of reductants in solid phase reactions should be reasonably controlled.

Xiao et al. prepared the black TiO_2_ by the solid-state chemical reduction strategy by mixing the sample with sodium borohydride in a certain proportion [87]. Then, the mixture was heated in a tubular furnace with N_2_ atmosphere [87]. Finally, the resulting sample was washed with deionized water to remove the unreacted sodium borohydride [87]. As shown in Figure 6, the color of the sample was getting darker with the temperature [87]. The absorption of the visible light was much enhanced after the reduction of TiO_2_ [87].

Zhu et al. showed that CaH_2_ can also be used as a constant reducing agent to prepare black TiO_2_ [88]. The reduction process was conducted at varied temperatures [88]. It was found that the obtained black TiO_2_ after reduction treatment at 400 °C had the best absorption of sunlight (over 80%), which was 11 times that of the pristine TiO_2_ [88]. This simple method provides an alternative for improving the absorption of visible light on the TiO_2_ surface.

Sinhamahapatra et al. reported the reduction of TiO_2_ particles to black TiO_2_ by magnesium thermal reduction method, which was inspired by the Kroll process, for the first time [89]. The synthetic procedure of this method was approximately identical to the method of sodium borohydride reduction [89]. TiO_2_ and magnesium powder were thoroughly mixed first, and then heated in a tube furnace at 650 °C with 5% H_2_/Ar for 5 h [89]. The obtained samples were placed in HCl solution for 24 h, and then washed with water to remove the acid, and finally dried at 80 °C [89].

### 3.3. Hot-Wire Annealing Method

In addition to the high temperature hydrogenation reduction and solid-phase reduction, researchers have also explored some other methods to synthesize black TiO_2_, which has made the method of preparing black TiO_2_ diversified. Wang et al. proposed a simple and direct hot-wire annealing (HWA) method [73]. The titanium dioxide nanorods were treated with highly active atomic hydrogen simply generated by hot wire [73]. The reduction mechanism was similar to that of the high temperature hydrogenation [73]. The resulted black TiO_2_ nanorods had better stability and higher photocurrent density compared with the traditional hydrogenation method [73]. In addition, it had no damage to the photoelectric chemical devices [73].

### 3.4. Anode Oxidation Method

The introduction of crystal defects to titanium dioxide can effectively extend the light absorption range to the visible light region without side effects. Anode oxidation is a simple and efficient method to synthesize defective black TiO_2_. Dong et al. successfully prepared black TiO_2_ using a two-step anode oxidation method [76]. The first step was to anodize Ti foil in the ethylene glycol solution with a certain proportion of NH_4_F and distilled water, and the corresponding voltage was set at 60 V [76]. After 10 h of oxidation, an oxide layer was obtained [76]. Subsequently, the Ti foil was purified to remove organic impurities, and treated at high temperature (450 °C) for 1 h to form black TiO_2_ [76].

### 3.5. Plasma Treatment

Zhu et al. prepared black TiO_2_ nanoparticles via the one-step solution plasma method under mild conditions [27]. The structural disorder layer was assumed to be formed in TiO_2_ after the solution plasma process [27]. The light absorption of TiO_2_ in the visible and near infrared range was significantly enhanced after the plasma treatment, thus increasing its activity in the water evaporation under solar illumination [27]. Teng et al. prepared black TiO_2_ using P25 as the precursor system, hydrogen plasma, and a hot filament chemical vapor deposition (HFCVD) device with H_2_ as the reducing gas [77]. The visible and near-infrared light absorption of TiO_2_ were much enhanced after the surface reduction [77]. Oxygen vacancies and Ti-H bonds were formed on the black TiO_2_ surface, thereby improving the photocatalytic activity [77]. 

### 3.6. Gel Combustion

Ullattil et al. prepared black anatase TiO_2−x_ photocatalysts through a one-pot gel combustion process using titanium butoxide, diethylene glycol, and water as precursors [90]. Plenty of Ti^3+^ and oxygen vacancies existed in the synthesized black anatase TiO_2_ nanocrystals confirmed by XPS measurements [90]. The light absorption of TiO_2_ was extended from UV to the near-infrared range [90]. Campbell et al. also synthesized black TiO_2_ via the sol-gel combustion method using titanium tetraisopropoxide as the precursor [91]. The light absorption ability was significantly enhanced compared to commercial TiO_2_ [91]. The obtained black TiO_2_ with the high surface area demonstrated much improved photocatalytic degradation efficiency of the organic dye under the visible light irradiation [91].

## 4. Strategies for Promoting Photocatalytic Activity of Black TiO_2_

Researchers have been trying to use metal and non-metal doping methods to prepare the modified TiO_2_ with better light absorption ability and photocatalytic activity. The introduction of metal ions and non-metallic elements into the TiO_2_ lattice can expand its absorption range to the visible light, thus enhancing the photocatalytic performance [92]. In recent years, doped black TiO_2_ has also been widely explored to narrow its band gap, thereby improving its optical properties in the visible light region, and enhancing its photocatalytic activity in various reactions.

### 4.1. Metallic Doped Black TiO_2_

It was found that by doping different metals in TiO_2_, Ti^4+^ in TiO_2_ lattice was replaced [93]. New impurity levels would be introduced in the band gap of TiO_2_ [93]. The band gap would be narrowed by the doping process, thus improving the separation efficiency of the photoelectron-hole of TiO_2_, increasing the quantum yield, and expanding the light absorption to the visible light region [92]. Photocatalytic degradation, hydrogen production capacity, and light energy conversion can be significantly improved [92]. Previously, various metal elements, including Cu, Co, Mn, Fe, Mo, etc., had been used to produce the doped TiO_2_ via different approaches [93]. Lately, some of the metal elements, such as Al, Ni, Na, etc., were also utilized to dope black TiO_2_ for achieving the narrower band gap and better photocatalytic performance [47,94,95].

Yi et al. prepared the amorphous Al-Ti-O nanostructure in black TiO_2_ via a scalable and low-cost strategy [94]. The commercial TiO_2_ and Al powders were mixed and then grinded in an agate mortar at room temperature for 0-50 min [94]. The color of the light gray TiO_2_ turned into gray after 2 min of milling [94]. Its color became much darker after the longer milling time [94]. Black Al-Ti-O oxide samples were obtained after milling for more than 5 min [94]. The color changes, UV-Vis-NIR diffuse reflectance spectra, and TEM image of the samples were shown in Figure 7 [94]. The crystalline Al and anatase TiO_2_ were transformed into amorphous Al-black TiO_2_ after the ball milling [94]. Al-black TiO_2_ after 20 min milling exhibited the best light absorption in the visible light and near infrared region [94]. 

Zhang et al. prepared Ni^2+^-doped porous black TiO_2_ photocatalysts through the combination of the sol-gel method and in situ solid-state chemical reduction process [95]. The reduction approach was performed by heating the mixture of Ni-doped TiO_2_ and NaBH_4_ at 350 °C under Ar atmosphere for 1 h [95]. The color of the white as-made TiO_2_ became yellowish after Ni doping [95]. Black Ni-doped TiO_2_ was obtained after the reduction with NaBH_4_ [95]. Figure 8 shows the optical properties and the band gap of different materials [95]. The light absorption of TiO_2_ was extended to the visible light range after Ni doping, and further enhanced after the chemical reduction, which was attributed to the generation of oxygen vacancies, Ti^3+^, and Ni^2+^ [95]. The band gap of the black Ni-doped TiO_2_ was only 1.96 eV [95].

Zhang et al. reported Ti^3+^ self-doped black TiO_2_ nanotubes with mesoporous nanosheet structure via a two-step approach consisting of the solvothermal reaction and hydrogenation process [82]. The appearance of the white TiO_2_ turned into black after hydrogenation at 600 °C for 2 h [82]. The optical absorption was significantly extended to the range of 400–800 nm after hydrogenation [82]. The band gap of pristine TiO_2_ decreased from 3.2 eV to 2.87 eV after the surface hydrogenation [82].

### 4.2. Non-Metallic Doped Black TiO_2_

The doping mechanism of nonmetallic elements can be explained as follows: the doped elements act as overlapping impurity levels in the valence band inside the photocatalyst crystal, thereby reducing the band gap of semiconductors and promoting the migration of photogenerated electrons to the active site. The doping of non-metal elements in the crystal lattice of TiO_2_ can slow down the electron-hole pair recombination rate, which is an effective modification way to improve the photocatalytic activity of TiO_2_.

The nitrogen atom, which has five outer shell electrons, has a similar radius to oxygen. The introduction of N into TiO_2_ enhances its visible light photocatalytic activity, which is proved to be the most ideal non-metallic doping element in a large number of studies [96,97]. Since the 2p orbital of N has a similar energy level to that of the oxygen atom and is easy to hybridize, the researchers found that the doping of N can improve the defects of TiO_2_ and broaden the response range of the absorption spectra [98]. The N-Ti-O bond generated by doping the crystal can change the energy level structure of TiO_2_ and improve the quantum efficiency [99]. In addition, N can also replace O in the lattice with the formation of the Ti-N bond, which increases the absorption of the visible light by TiO_2_ and improves the photocatalytic efficiency of TiO_2_ [96].

Zhou et al. successfully synthesized the nitrogen-doped black titanium dioxide nanocatalyst by calcining white TiO_2_ with or without urea at varied temperatures under the different atmosphere [31]. The N-doped TiO_2_ using urea as N precursor has a better visible light absorption, narrower band gap and the most effective excitation charge separation, and higher photocatalytic activity [31]. Liu et al. prepared N-doped black TiO_2_ spheres via a two-step process consisting of the solvothermal reaction and calcination in the nitrogen atmosphere [100]. The black N-TiO_2_ photocatalysts were obtained after heat treatment at 500 °C in N_2_ atmosphere for 3 h [100]. Ammonium chloride was used as the nitrogen source during the synthetic process [100]. The obtained black N-TiO_2_ with a moderate mole ratio of ammonium chloride to TiO_2_ (2:1) had the narrowest band gap and the highest photocatalytic pollutant degradation efficiency [100].

Gao et al. prepared black TiO_2_ nanotube arrays with dual defects consisting of bulk N doping and surface oxygen vacancies [101]. Urea was utilized as the N precursor during the anodic oxidation process [101]. Black N-doped TiO_2_ nanotube arrays were obtained after calcination at 600 °C in the Ar atmosphere using aluminum powder [101]. The doping of N generated a new energy level and shortened the carrier migration distance [101]. The synergistic effect of the two defects established an internal electric field, promoted the transfer of charge, and achieved the balance between kinetics and thermodynamics, thereby enhancing the photocatalytic hydrogen production efficiency [101].

Cao et al. successfully synthesized N and Ti^3+^ co-doped mesoporous black TiO_2_ hollow spheres (N-TiO_2−x_) by a step-by-step method [102]. The prepared three different kinds of TiO_2_ had similar XRD peaks, indicating no impurities was formed during the synthetic process, as presented in Figure 9a [102]. The broader peaks of the black TiO_2_ may be attributed to the lattice distortion caused by N doping [102]. Raman spectra (Figure 9b) showed that the main phase of the titanium dioxide hollow sphere was anatase [102]. The co-doping of Ti^3+^ and N in N-TiO_2−x_ resulted in a certain amount of attenuation [102]. The absorption in the visible light was much enhanced after the co-doping of N and Ti^3+^ in TiO_2_ [102]. The color of the samples gradually became darker after the nitrogen and Ti^3+^ doping [102]. In addition, the band gap of the black N-doped TiO_2_ was much smaller than the pristine TiO_2_ (Figure 9c,d) [102]. The charge separation and photocatalytic activity for photocatalytic pollutant removal and hydrogen generation were much improved after the co-doping of N and Ti^3+^ in the lattice of TiO_2_ (Figure 9e) [102]. 

### 4.3. Metal-Loaded Black TiO_2_

Metal nanoparticles, such as Ag, Cu, Pt, etc., can generate the surface plasmon resonance (SPR) effect, thus improving the UV-vis absorption ability of photocatalysts [103]. The introduction of metal nanoparticles or clusters to the black TiO_2_ photocatalyst surface could further expand its light absorption range and enhance the photo-induced charge separation and transfer efficiency, thus improving the photocatalytic performance [38,104,105,106,107,108,109]. Silver nanoparticles or clusters have been extensively explored to construct Ag/black TiO_2_ photocatalysts due to the relatively lower cost of Ag than other noble metals and the SPR effect. Jiang et al. prepared the Ag-decorated 3D urchinlike N-TiO_2-x_ via a facile photo-deposition method combined with a reduction process, as presented in Figure 10 [104]. The AgNO_3_ solution was used as the Ag precursor and deposited onto the N-TiO_2_ surface under UV illumination at the wavelength of 365 nm for 30 min [104]. Notably, the unique 3D urchinlike structure was retained after the Ag deposition and NaBH_4_ reduction process [104]. The light absorption of photocatalysts in the visible light range was further enhanced after the Ag deposition, with a much smaller band gap (2.61 eV) [104]. The Ag/N-TiO_2−x_ photocatalysts presented the most excellent photocatalytic H_2_ production rate (186.2 μmol h^−1^ g^−1^) [104].

Li et al. constructed Ag nanoparticle-decorated black TiO_2_ foams through the wet impregnation and high temperature surface hydrogenation process [105]. Ag nanoparticles were formed in the open pores of black TiO_2_ foams after the hydrogen atmosphere reduction, thus decreasing its surface area [105]. The synthetic process and UV-vis absorption spectra of Ag-black TiO_2_ foams were presented in Figure 11 [105]. The Ag-black TiO_2_ foams with varied amounts of silver showed apparent absorption at around 500 nm thanks to the SPR effect of Ag nanoparticles [105]. The Ag-black TiO_2_ foams containing 3 wt.% Ag nanoparticles exhibit the highest photocatalytic efficiency for atrazine removal [105]. Excess amounts of Ag nanoparticles in black TiO_2_ foams would decrease its photocatalytic performance due to the aggregation of Ag nanoparticles [105]. Ag nanoparticles were also decorated onto black TiO_2_ nanorods [106,109] and nanotubes surface [38], to further improve its photocatalytic performance using the SPR effect. In addition, NiS and Pt nanoparticles were co-decorated onto the surface of black TiO_2_ nanotubes via the solvothermal and photo-deposition approach, respectively [109]. The SPR effect of Pt nanoparticles effectively improved the light absorption ability, thus enhancing the photocatalytic water splitting [109]. Wang et al. successfully deposited Pt single atoms onto the black TiO_2−x_/Cu_x_O surface assisted by the presence of surface oxygen vacancies [110]. The deposition of Pt single atoms further improved the light absorption of black TiO_2−x_/Cu_x_O in the entire visible region [110]. Cu, which was a much cheaper candidate than noble metal, was also used for the surface decoration of black TiO_2_ surface as a SPR effect metal, thereby improving its photothermal effect [108]. 

### 4.4. Construction of Black TiO_2_ Based Heterojunction Photocatalysts

Although the visible light absorption ability of TiO_2_ has been much enhanced after the hydrogenation or reduction process, the charge separation and transfer efficiency of black TiO_2_ is still far from satisfactory for photocatalytic applications. In addition to the surface modification with metal nanoparticles, the construction of black TiO_2_-based junctions is an efficient way to improve the photo-generated charge separation and migration efficiency. The types of black TiO_2_-based heterojunctions can be divided into three main categories, including type II heterojunctions [59,111,112,113,114,115,116,117,118,119], Z-scheme heterojunctions [120], and tandem heterojunctions [29,121,122]. Tan et al. fabricated the Ti^3+^-TiO_2_/g-C_3_N_4_ nanosheets heterojunctions through a facile calcinations-sonication assisted approach [118]. The photo absorption of Ti^3+^-TiO_2_ in the visible light range had been evidently enhanced after the coupling with meso-g-C_3_N_4_ [118]. The synthetic procedure of Ti^3+^-TiO_2_/g-C_3_N_4_ nanosheets, the UV-vis absorption spectra, and the band gaps of different samples were shown in Figure 12 [118]. The separation and migration of the photo-induced charge carrier had been effectively improved due to the construction of heterojunctions [118]. Ti^3+^-TiO_2_/g-C_3_N_4_ nanosheets exhibited the highest photocatalytic H_2_ evolution rate and phenol degradation efficiency [118]. A type II heterojunction with enhanced charge separation and transfer efficiency had been proposed [118]. 

Ren et al. prepared magnetic γ-Fe_2_O_3_/black TiO_2_ heterojunctions via the metal-ion intervened hydrothermal method and high temperature hydrogenation process [59]. In addition, α-Fe_2_O_3_ nanosheets were transformed to the surface defected γ-Fe_2_O_3_ after the hydrogenation process [59]. The light utilization of black TiO_2_ in visible light even near the infrared region has been improved after the combination with γ-Fe_2_O_3_ [59]. Figure 13 presented the UV-vis absorption spectra and the proposed band structure and charge transfer mechanism of γ-Fe_2_O_3_/black TiO_2_ heterojunctions [59]. The fabrication of the type II heterojunctions efficiently enhanced the photo-generated charge separation and transfer process [59]. The photocatalytic degradation of tetracycline on the heterojunction photocatalysts surface had been much improved, compared with the pristine TiO_2_ [59]. In addition, Bi_2_MoO_6_ [111], CdS [115], CeO_2_ [117], SrTiO_3_ [119], etc. had also been used for the construction of type II heterojunctions with black TiO_2_ with much improved photocatalytic efficiency.

Sun et al. synthesized the CdS quantum dots/defective ZnO_1−x_-TiO_2−x_ Z scheme heterojunction via the combination of hydrothermal synthesis, chemical reduction, and electroless planting process [120]. The visible light absorption was enhanced by the formation of ZnO-TiO_2_ heterojunction, and further improved after combining with CdS [120]. The formation process of the heterojunction, UV-vis spectra, and proposed charge transfer mechanism are shown in Figure 14 [120]. The charge separation efficiency and photocatalytic organic pollutant removal rate were much improved upon the formation of CdS QDs/defective ZnO_1−x_-TiO_2−x_ Z scheme heterojunction [120].

To further promote the visible light utilization and photo-induced charge separation and transfer efficiency, black TiO_2_-based tandem heterojunction photocatalysts have been proposed by researchers for photocatalytic hydrogen production. Sun et al. prepared a hierarchical hollow black TiO_2_/MoS_2_/CdS tandem heterojunction photocatalyst through the combination of the solvothermal method and high-temperature hydrogenation treatment [122]. The black TiO_2_/MoS_2_ heterojunction effectively enhanced the photon absorption in visible light and the near infrared region [122]. The tandem system further promoted visible light utilization compared to other combinations [122]. The schematic view of the construction of black TiO_2_/MoS_2_/CdS, UV-vis absorption spectra, and photocatalytic hydrogen evolution rate are illustrated in Figure 15. The charge separation and migration efficiency were much promoted by the formation of black TiO_2_/MoS_2_/CdS tandem heterojunction [122]. The tandem heterojunction exhibited the highest photocatalytic hydrogen production rate under AM 1.5 illumination [122]. In addition, a sandwich-like mesoporous black TiO_2_/MoS_2_/black TiO_2_ nanosheet photocatalyst was proposed for visible light photocatalytic hydrogen generation with a much-promoted photo-generated charge transfer efficiency [121]. The mesoporous TiO_2_ and Cu_2_S were also combined with MoS_2_ to synthesize the hierarchical tandem heterojunctions [29]. The near-infrared energy utilization was enhanced by the tandem system, thereby promoting the photothermal effect [29]. The visible light photocatalytic H_2_ generation rate was significantly improved, achieving 3376.7 μmol h^−1^ g^−1^, which was approximately 16 times that of black TiO_2_ [29].

## 5. Applications of Black TiO_2_

The absorption of light can be extended from the ultraviolet light to visible light, and even to near-infrared light by changing the white TiO_2_ into black TiO_2_ via various methods. This strategy can be utilized for enhancing the photocatalytic activity in the visible light range. Changing the color of TiO_2_ from white to black is one of most efficient ways for improving its photocatalytic efficiency in various fields, such as photocatalytic water splitting, photocatalytic pollutant degradation, etc.

### 5.1. Photocatalytic Water Splitting

Black TiO_2_ has a modified band structure, thereby improving the charge separation and migration efficiency. Its enhanced photocatalytic performance has been extensively investigated in water splitting. Black mesoporous TiO_2_ synthesized by Zhou et al. has excellent hydrogen production performance [79]. As shown in Figure 16A, black TiO_2_ had a higher rate of hydrogen production than original TiO_2_ under the condition of AM 1.5G and had excellent hydrogen production ability in visible light [79]. In addition, almost no attenuation was detected during photocatalytic measurements after 10 cycles [79]. As presented in Figure 16B, its apparent quantum efficiency at each single wavelength was much higher than that of the original sample [79]. The mesoporous black TiO_2_ photocatalysts showed remarkable photocatalytic stability in 10 cycling hydrogen evolution measurements within 30 h (each cycling test was conducted in the presence of fresh 1 mL methanol) [79]. The black TiO_2_ nanotubes prepared by Yang et al. showed an excellent photocatalytic hydrogen production performance (9.8 mmol h^−1^g^−1^) through high temperature hydrogenation [80]. The enhanced photocatalytic activity could be attributed to two aspects: (1) the special one-dimensional hollow tube structure improved the charge separation efficiency; (2) the high temperature hydrogenation strategy improved its ability for sunlight utilization [80]. The photocatalytic activity of the black TiO_2_ nanotubes for H_2_ production remained stable for 5 cycles in 15 h using H_2_PtCl_6_ as the co-catalyst and methanol as the sacrificing agent [80]. 

Two-dimensional lamellar structures with plenty of active sites are often used in TiO_2_ photocatalysis because of their large specific surface area. The black TiO_2_ nanosheets prepared by Zhang et al. shortened the band gap to 2.85 eV, thereby broadening the light response to the visible light region [123]. The hydrogen production rate was up to 165 μmol h^−1^ 0.05 g^−1^, which was twice as much as that of the original sample [123]. The chemical stability, light corrosion resistance, and photocatalytic activity for H_2_ generation were confirmed by 5 cycling tests in 25 h, using H_2_PtCl_6_ and methanol as the co-catalyst and sacrificing agent, respectively [123]. Similarly, Wu et al. synthesized another black TiO_2_ nanosheets, which had a high hydrogen production rate of 3.73 mmol h^−1^ g^−1^ [83]. This photocatalyst exhibited an unchanged photocatalytic H_2_ evolution rate in 5 cycling measurements with 15 h [83]. Li et al. designed and synthesized black TiO_2_ nanospheres by the self-assembly solvothermal method combined with the hydrogenation strategy [81]. The charge separation efficiency had been effectively improved after the hydrogenation process, confirmed by experimental results and DFT calculations [81]. The photocatalytic performance for H_2_ formation was also repeated 5 times in 15 h and remained stable in the 5 cycles [81].

The black rutile TiO_2_ prepared by Xiao et al. by the solid-phase reduction method showed much-enhanced hydrogen production performance, stability, and high apparent quantum efficiency, which was about 1.5 times that of the original sample (Figure 17) [87]. In addition, the defects in TiO_2_ could be regulated by varying the hydrogenation temperature, and the optimal hydrogenation temperature was proved to be 300 °C [87]. The stability of photocatalytic activity was verified by 5 cycling hydrogen formation measurements in 12 h under AM 1.5 illumination [87]. No obvious decrease in the activity was observed in the cycling tests [87]. The maximum hydrogen production rate of black TiO_2_ synthesized by Sinhamahapatra et al. using the controllable magnesium thermal reduction method was 43 mmol h^-1^ g^-1^ in the full solar wavelength range with excellent stability, which was better than the black TiO_2_ material previously reported [89]. The black TiO_2_ nanoparticles presented great stability in the photocatalytic hydrogen evolution confirmed by 10 cycling measurements, which were conducted for 10 consecutive days using the same solution [89]. The aging of the black TiO_2_ materials was generally not mentioned in the reported publications. The photocatalytic stability of the hydrogen generation was mostly measured in 5 cycling tests within 15 h using the same solution. Some photocatalysts were tested in 10 repeated photocatalytic hydrogen formation measurements for more than 20 h. Black TiO_2_ photocatalysts usually showed good stability and light corrosion resistance in the photocatalytic H_2_ evolution reaction, providing the possibility of long-term usage of black TiO_2_ photocatalysts for H_2_ production.

### 5.2. Photocatalytic Degradation of Pollutants

In addition to the hydrogen production, pollutant degradation is also one of the main applications of photocatalysis. Titanium dioxide was often used in the photocatalytic degradation of organic dye and pesticides [124,125,126]. Black TiO_2_ with an enhanced light absorption ability would have much improved the photocatalytic pollutant removal efficiency. The black TiO_2_ obtained by CaH_2_ reduction not only presented an enhanced hydrogen generation rate, which was 1.7 times that of the original sample, but also achieved a huge improvement in the degradation of pollutants with the complete removal of methyl orange within 8 min [72]. Hamad et al. synthesized black TiO_2_ using a new method of controlled hydrolysis [85]. The oxygen vacancy concentration was significantly increased with a much-reduced band gap, thereby showing an excellent organic pollutant degradation rate under visible light irradiation [85]. 

The oxygen vacancy plays an important role in photocatalysis. Black TiO_2_ prepared by Teng et al. via vapor deposition had a high photocatalytic oxidation activity for organic pollutants in the water, due to the formation of Ti-H bonds and a large number of oxygen vacancies [77]. All pollutants (rhodamine B) could be completely degraded within 50 min detected by the UV-vis spectrophotometer [77]. The defective TiO_2−x_ prepared by the anodic oxidation method was characterized by the electron paramagnetic resonance spectroscopy, confirming the existence of oxygen vacancies and the extension of the absorption from the ultraviolet to visible light region [76]. This black TiO_2_ material showed excellent photocatalytic degradation activity for rhodamine B under 400–500 nm light irradiation [76]. 

The black TiO_2_-based heterojunction could significantly improve its photocatalytic efficiency in pollutant remediation thanks to the enhanced charge separation and transfer efficiency. Jiang et al. prepared black TiO_2_/Cu_2_O/Cu composites via in-situ photodeposition and the solid reduction method [127]. The light energy harvesting in the visible and infrared range was much enhanced after the formation of the composites [127]. The photocatalytic efficiency of the composites for Rhodamine B degradation was improved compared with the commercial P25, due to the enhanced charge separation efficiency [127]. Qiang et al. synthesized the RuTe_2_/black TiO_2_ photocatalyst through gel calcination and the microwave-assisted process [128]. The light absorption range of the as-made composites was enlarged compared to the pristine TiO_2_. The photocatalytic efficiency of the diclofenac degradation was 1.2 times higher than the pure black TiO_2_ [128]. The stability of RuTe_2_/black TiO_2_ for the photocatalytic diclofenac degradation was confirmed via 5 repeated experiments [128].

Tetracycline is a toxic antibiotic which is difficult to remove. Li et al. synthesized black TiO_2_ modified with Ag/La presented an improved visible light photocatalytic performance for the tetracycline degradation [26]. The photocatalytic stability and reusability of black TiO_2_-based photocatalysts were studied via 5 cycling tests without apparent deactivation [26]. Wu et al. reported that the synthesized black anatase TiO_2_ exhibited impressive photocatalytic degradation of tetracycline [84]. Its degradation efficiency of tetracycline was 66.2% under the visible light illumination, which was higher than that of the white titanium dioxide and doped titanium dioxide [84]. In addition, ·O^2−^ and h^+^ were found to play important roles in the degradation process, which was different from the original TiO_2_, providing new insights for environmental protection [84]. The stability of the photocatalytic tetracycline degradation was measured in four repeated experiments within 960 min without apparent deactivation after four cycles [84]. Table 2 summarizes the applications and photocatalytic stability of the black TiO_2_ nanomaterials. The long-term photocatalytic stability of pollutant removal is often overlooked and unverified in most reported research. Therefore, researchers should pay more attention to aging and photocatalytic stability in pollutant degradation in the future. 

## 6. Summary and Outlooks

The utilization and conversion of sunlight in a more efficient way has gained much interest due to the energy crisis and global warming effect. The construction of black TiO_2_-based materials is proved to be an effective approach for promoting visible light utilization. Black TiO_2_ with various morphologies, such as nanospheres, nanotubes, nanowires, etc., have been rationally designed. The properties of black TiO_2_, such as the surface area, are easily affected by its morphology. Although the photon absorption of TiO_2_ in the visible light region can be effectively increased after the surface reduction process, its photocatalytic performance still needs to be improved for practical applications. The photocatalytic activity of black TiO_2_ photocatalysts can be further enhanced by three main methods: element doping, decoration with metal nanoparticles, and fabrication of heterojunctions. The introduction of metal or nonmetal elements into the black TiO_2_ lattice can create new impurities, thus narrowing its band gap. The SPR effect caused by metal decoration on the black TiO_2_ surface can efficiently improve its visible light utilization. The fabrication of black TiO_2_ heterojunctions, including type II, Z scheme, and tandem heterojunctions, can significantly enhance the photo-induced charge separation and transfer efficiency, thereby promoting the photocatalytic performance. The photocatalytic activity of black TiO_2_-based materials are mainly evaluated in the photocatalytic hydrogen production and pollutant removal. Although these black TiO_2_-based nanomaterials exhibit excellent photocatalytic activity in the visible light region, technologies for enhancing light harvesting in near-infrared should be developed. In addition, the enhancement of the photo-generated charge separation and transfer should be further reinforced to meet the standard for practical application. Applications of black TiO_2_-based materials in industrial and outdoor fields, such as self-cleaning surfaces, should also be investigated in future.

## Figures and Tables

**Figure 1 nanomaterials-13-00468-f001:**
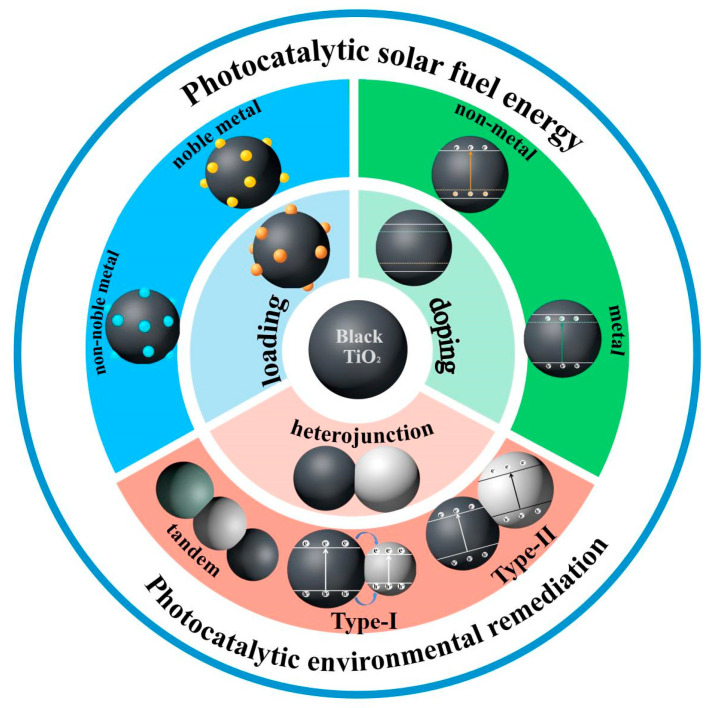
Outline diagram of the types of black TiO_2_ nanomaterials.

**Figure 3 nanomaterials-13-00468-f003:**
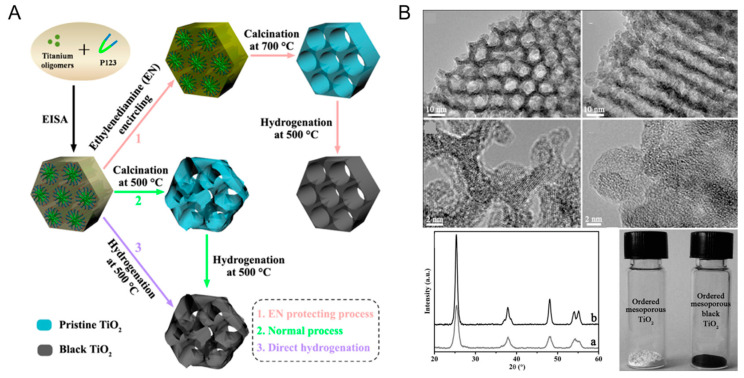
(**A**) Schematic synthesis process for the ordered mesoporous black TiO_2_ materials. (**B**) Representative TEM images along [100] and [110] planes, HRTEM images of the ordered mesoporous black TiO_2_ materials, and X-ray diffraction patterns and the photos of the ordered mesoporous black TiO_2_ materials (a) and ordered mesoporous TiO_2_ materials (b). Reprinted with permission for ref. [79]. Copyright 2014, American Chemical Society.

**Figure 4 nanomaterials-13-00468-f004:**
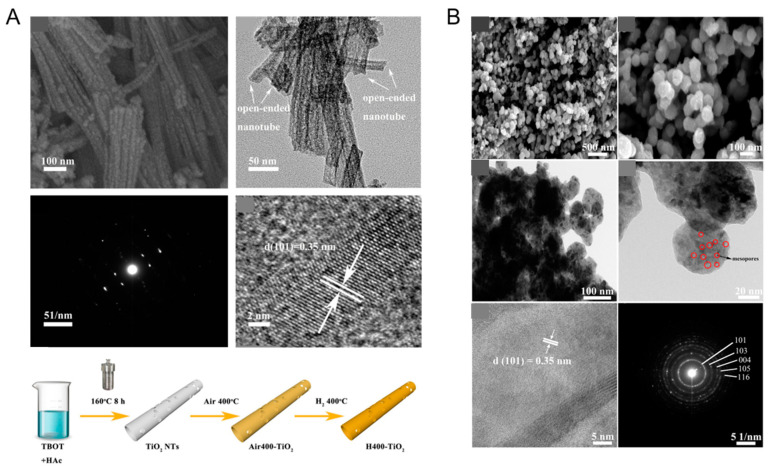
(**A**) SEM, TEM, SAED, HRTEM images and schematic illustration of the formation process of H400-TiO_2_. Reprinted with permission for ref. [80]. Copyright 2021, Elsevier. (**B**) Representative SEM, TEM, HRTEM images and the corresponding elected area electron diffraction (SAED) pattern of the mesoporous TiO_2_ nanospheres after surface hydrogenation. Reprinted with permission for ref. [81]. Copyright 2021, Elsevier.

**Figure 5 nanomaterials-13-00468-f005:**
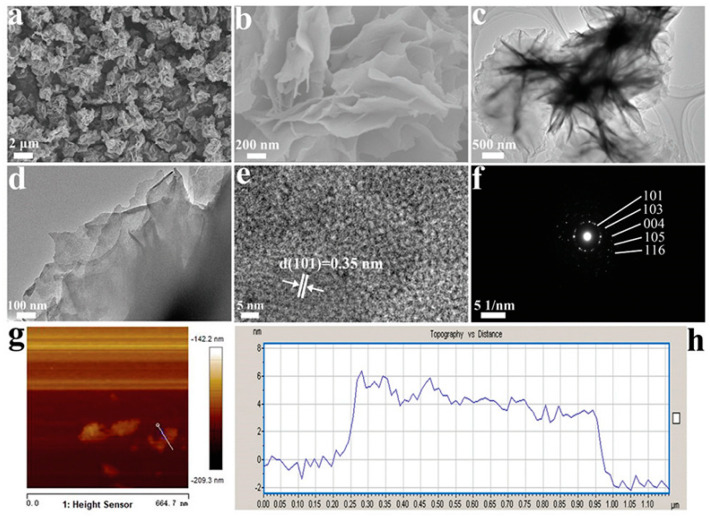
(**a**,**b**) SEM images, (**c**,**d**) TEM images, (**e**) HRTEM, (**f**) SAED pattern, (**g**) AFM topography image, and (**h**) the corresponding height information of the 2D ultrathin nanosheets. Reprinted with permission for ref. [83]. Copyright 2020, The Royal Society of Chemistry.

**Figure 6 nanomaterials-13-00468-f006:**
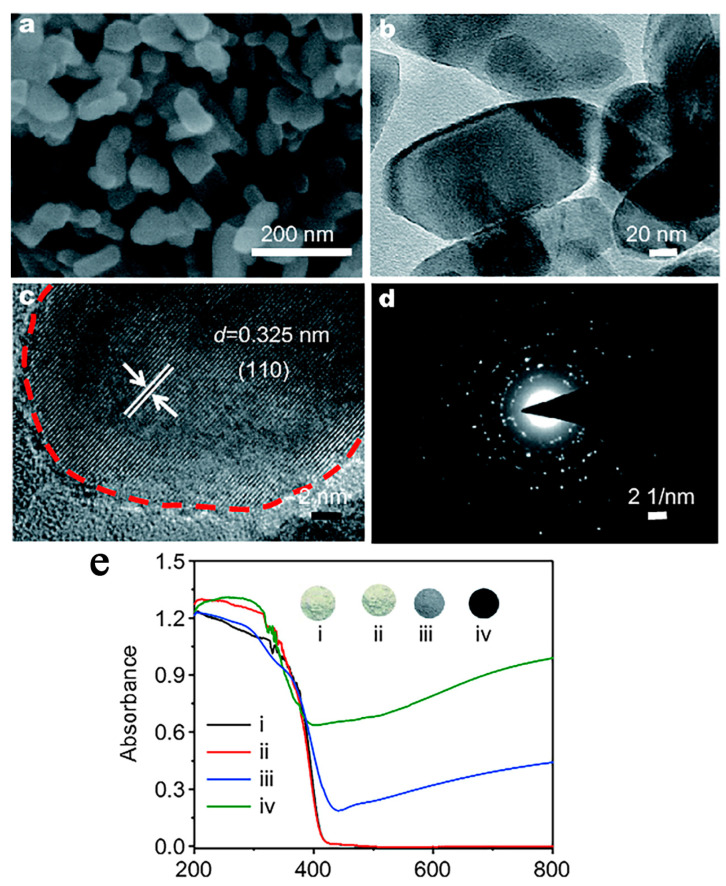
SEM (**a**), TEM (**b**), HRTEM images (**c**), the corresponding selected-area electron diffraction pattern (**d**) of the hydrogenated rutile TiO_2_ (300 °C) and the UV/vis absorption spectra (**e**) of the pristine rutile TiO_2_ (i) and the hydrogenated rutile TiO_2_ under 250 °C (ii), 300 °C (iii), and 350 °C (iv). Reprinted with permission for ref. [87]. Copyright 2018, Springer.

**Figure 7 nanomaterials-13-00468-f007:**
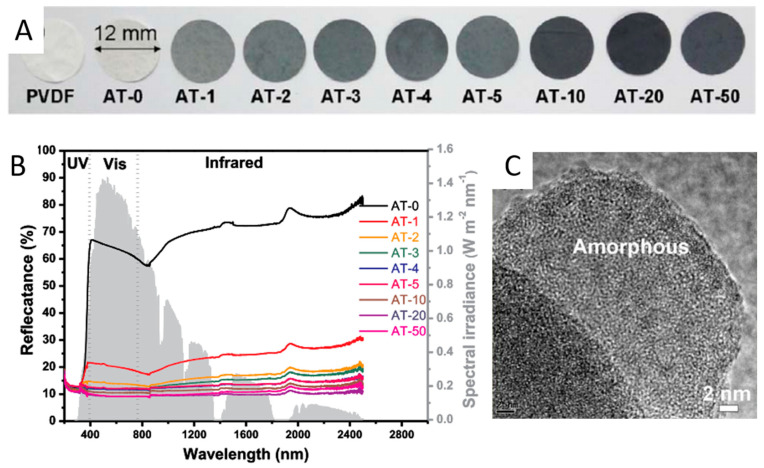
(**A**) the set of the AT-x (Al-Ti-O-milling time) membranes; (**B**) UV-Vis-NIR diffuse reflectance spectra of AT-x; (**C**) TEM image of Al-black TiO_2_. Reprinted with permission for ref. [94]. Copyright 2017, Elsevier.

**Figure 8 nanomaterials-13-00468-f008:**
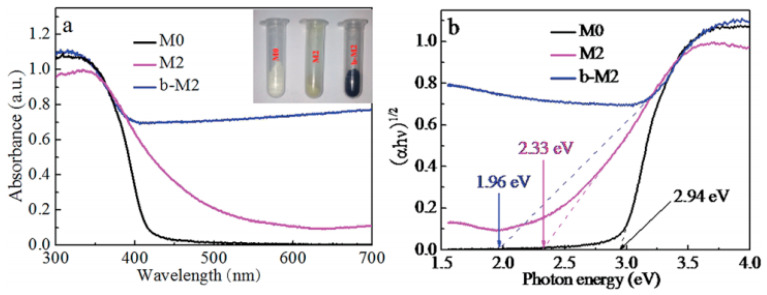
(**a**) UV-vis absorption spectra of TiO_2_ (M0), Ni-doped TiO_2_ (M2), and black Ni-TiO_2_ (b-M2); (**b**) band gap for M0, M2, and b-M2 samples, respectively. Reprinted with permission for ref. [95]. Copyright 2015, The Royal Society of Chemistry.

**Figure 9 nanomaterials-13-00468-f009:**
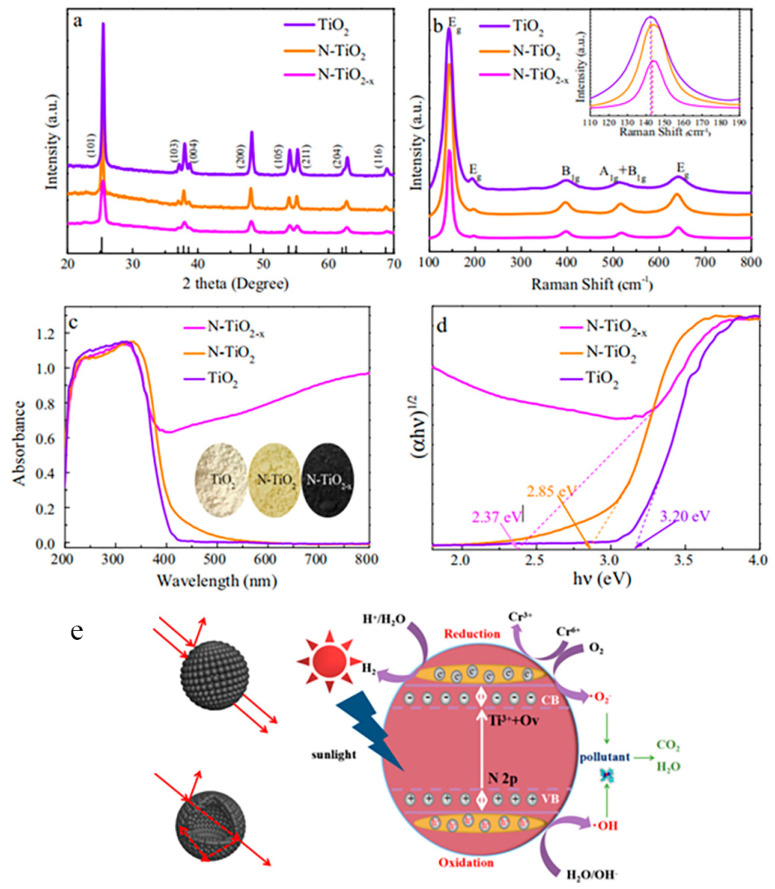
XRD patterns (**a**), Raman spectra (**b**), UV-vis adsorption spectra (**c**), and determination of the indirect interband transition energies (**d**) of TiO_2_, N-TiO_2_, and N-TiO_2−x_, respectively. Schematic diagrams of light pathways in nanoparticles, hollow spheres, and schematic illustration for the solar-driven photocatalytic mechanism of N-TiO_2−x_ (**e**). Reprinted with permission for ref. [102]. Copyright 2017, Elsevier.

**Figure 10 nanomaterials-13-00468-f010:**
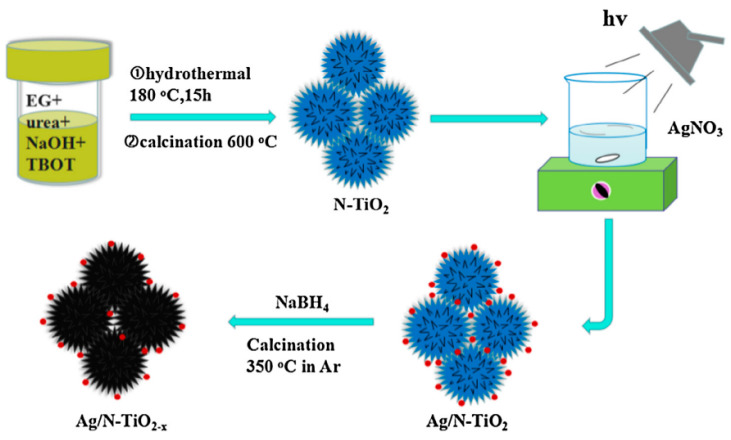
Schematic illustration for preparation of 3D urchinlike Ag/N-TiO_2−x_. Reprinted with permission for ref. [104]. Copyright 2018, Elsevier.

**Figure 11 nanomaterials-13-00468-f011:**
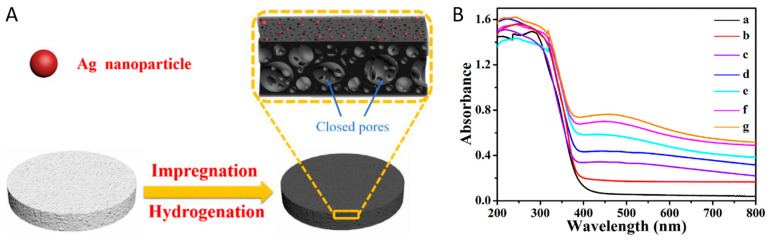
(**A**) Schematic view for the synthetic process of Ag/black TiO_2_ foams; (**B**) The UV-vis absorption spectra of black TiO_2_ foams (a) and Ag-black TiO_2_ foams with different Ag contents of 0.5 (b), 1 (c), 2 (d), 3 (e), 4 (f), and 5 wt.% (g). Reprinted with permission for ref. [105]. Copyright 2018, Elsevier.

**Figure 12 nanomaterials-13-00468-f012:**
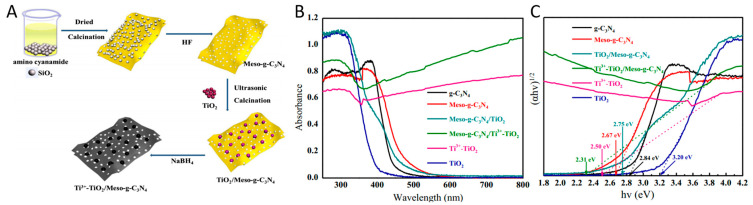
(**A**)The synthetic process of Ti^3+^-TiO_2_/g-C_3_N_4_ nanosheets heterojunctions; (**B**) UV-vis diffuse reflectance spectra; (**C**) the corresponding calculated band gaps. Reprinted with permission for ref. [118]. Copyright 2018, Elsevier.

**Figure 13 nanomaterials-13-00468-f013:**
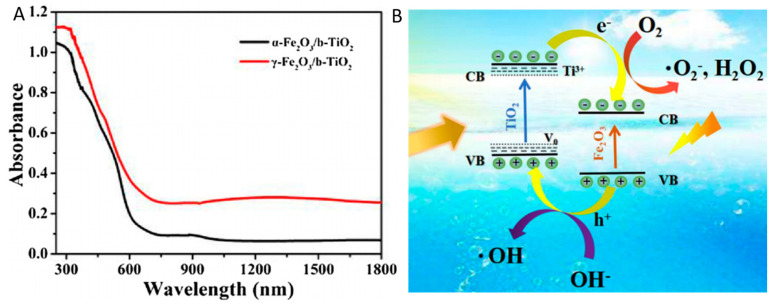
(**A**) UV-vis spectra; (**B**) Proposed band alignment and charge transfer mechanism. Reprinted with permission for ref. [59]. Copyright 2019, Elsevier.

**Figure 14 nanomaterials-13-00468-f014:**
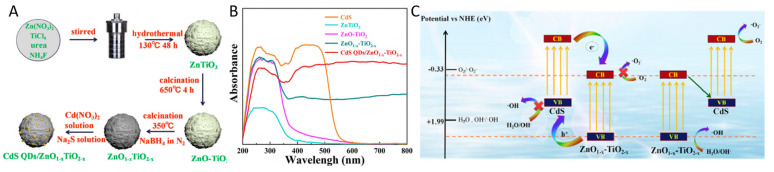
(**A**) Formation process of CdS QDs/ZnO_1−x_-TiO_2−x_; (**B**) UV-vis spectra; (**C**) Proposed band alignment and charge transfer mechanism. Reprinted with permission for ref. [120]. Copyright 2020, Elsevier.

**Figure 15 nanomaterials-13-00468-f015:**
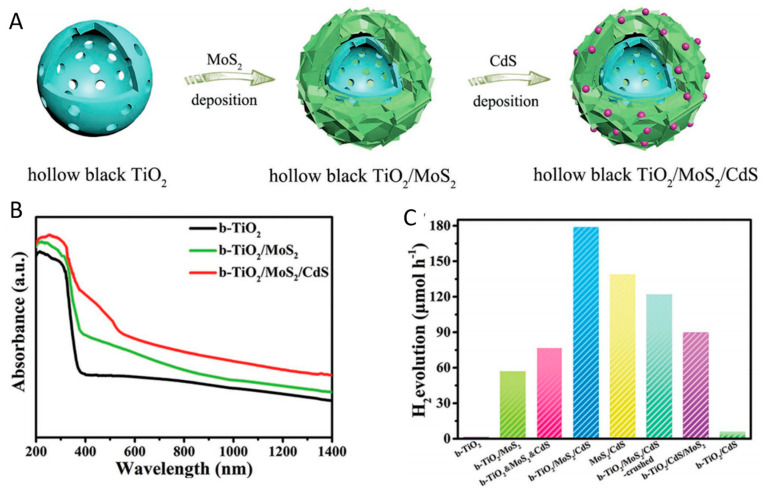
(**A**) Schematic illustration for the formation of TiO_2_/MoS_2_/CdS, (**B**) UV-vis spectra, (**C**) photocatalytic H_2_ evolution measurements. Reprinted with permission for ref. [122]. Copyright 2018, Wiley-VCH.

**Figure 16 nanomaterials-13-00468-f016:**
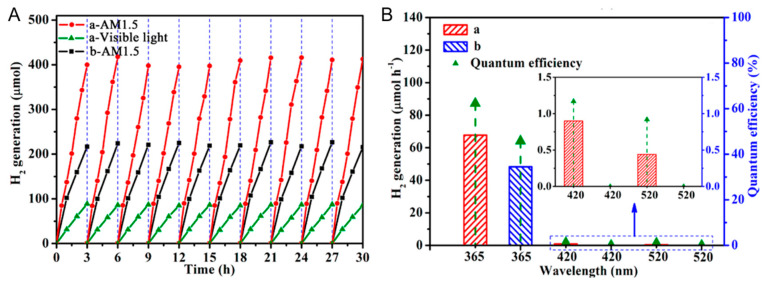
Photocatalytic hydrogen evolution of the ordered mesoporous black TiO_2_ (a) and pristine ordered mesoporous TiO_2_ materials (b). (**A**) Cycling tests of photocatalytic hydrogen generation under AM 1.5 and visible light irradiation. (**B**) The photocatalytic hydrogen evolution rates under single-wavelength light and the corresponding QE. The inset enlarges the QE of single-wavelength light at 420 and 520 nm. Reprinted with permission for ref. [79]. Copyright 2014, American Chemical Society.

**Figure 17 nanomaterials-13-00468-f017:**
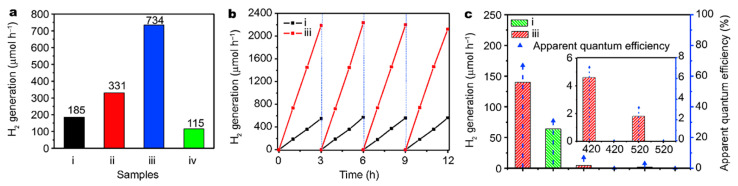
(**a**) Photocatalytic hydrogen evolution rates, (**b**) cycling tests of photocatalytic hydrogen generation under AM 1.5 irradiation, (**c**) the single wavelength photocatalytic hydrogen evolution rates of pristine TiO_2_ (i), and hydrogenated TiO_2_ at 250 °C (ii), 300 °C (iii), and 350 °C (iv), respectively. Reprinted with permission for ref. [87]. Copyright 2018, Springer.

**Table 1 nanomaterials-13-00468-t001:** Properties of black TiO_2_ nanomaterials.

Crystal Phase	Reducing Agent	Color	Morphology	Oxygen Vacancies	Ref.
Anatase with minor rutile	NaBH_4_	Black	Nanospheres	✔	[26]
Anatase and rutile	Hydrogen radicals	Black	Nanospheres	✔	[27]
Anatase	NaBH_4_	Black	Nanoparticles	✔	[28]
Anatase and rutile	Hydrogen	Black	Microspheres	✔	[29]
Anatase and rutile	PulsedLaser Ablation in Liquid	Black	Core-shell microspheres	✔	[30]
Anatase	Urea	Black	Nanoparticles	✔	[31]
Anatase	NaBH_4_	Black	Nanospheres	✔	[32]
Anatase	NaBH_4_	Black	Nanospheres	✔	[33]
Anatase	NaBH_4_	Black	Nanospheres	✔	[34]
Anatase	Electrochemical reduction	Black	Nanotubes	✔	[35]
Anatase	CaH_2_	Black	Nanotubes	✖	[36]
Anatase	Hydrogen	Black	Nanotubes	✔	[37]
Anatase	Ag particles	Black	Nanotubes	✔	[38]
Anatase with minor rutile	Electrochemical reduction	Black	Nanotubes	✔	[39]
Anatase and rutile	Electrochemical reduction	Blue and black	Nanotube arrays	N/S	[40]
Anatase and rutile	Electrochemical reduction	Dark yellow	Nanotube arrays	✔	[41]
Anatase and rutile	aluminothermic reduction	Black	Nanotubes	✔	[42]
Anatase	Electrochemical reduction	Black	Nanotube arrays	✔	[43]
Anatase	Hydrogen	Black	Nanowires	✔	[44]
Anatase	NaBH_4_	Black	Nanowires	✔	[45]
Anatase	Glycerol	Black	Nanoparticles and nanowires	✔	[46]
Rutile	NaBH_4_	N/S	Hollow nanowires	✔	[47]
Anatase	H_2_S and SO_2_	Black	Nanoplatelets	✔	[48]
Anatase	NaBH_4_	Black	Nanoplates	✔	[49]
Anatase	Hydrogen	Black	Nanosheets	✔	[50]
Anatase	Aluminothermic reduction	Black	Nanosheet array films	✔	[51]
Anatase	Hydrogen	Black	Nanobelts	✔	[52]
Anatase	NaBH_4_	Black	Nanobelts	✔	[53]
Anatase	Hydrogen	Black	Nanocages	✔	[54]
Anatase	Hydrogen	Black	3D nanoflower	✔	[55]
Anatase and rutile	NaBH_4_	Black	Nanofiber	✔	[56]
Anatase	NaBH_4_	Black	Hollow shell	✔	[57]
Anatase	Annealing in vacuum	Black	Core-shell	✔	[58]
Anatase	Hydrogen	Black	Mesoporus hollow sphere	✔	[59]
Anatase and rutile	Electrochemical	Black	Films	✔	[60]
Anatase	H_2_ plasma treatment	Black	Mesoporous films	✔	[61]
Anatase	Hydrogen	Black	Hierarchical nanolace films	✔	[62]

Notes: N/S: not studied; ✔: yes; ✖: no.

**Table 2 nanomaterials-13-00468-t002:** Applications and properties of black TiO_2_ nanomaterials.

Black TiO_2_	Reducing Agent	Oxygen Vacancy	Application	Numbers of Cycling Tests	Ref.
Anatase with minor rutile	NaBH_4_	✔	Tetracycline degradation	5	[26]
Anatase and rutile	H_2_	✔	Rhodamine B degradation	N/S	[77]
Anatase	H_2_	✔	H_2_ evolution	10	[79]
Anatase	H_2_	✔	H_2_ evolution	5	[80]
Anatase	H_2_	✔	H_2_ evolution	5	[81]
Anatase	H_2_	✔	H_2_ evolution	5	[83]
Anatase	H_2_	✔	Tetracycline degradation	4	[84]
Anatase	H_2_	N/S	Orange G degradation	N/S	[85]
Rutile	NaBH_4_	✔	H_2_ evolution	4	[87]
Anatase with minor rutile	H_2_	✔	H_2_ evolution	10	[89]
Anatase	Anodization	✔	Rhodamine B degradation	N/S	[76]
Anatase	H_2_	N/S	H_2_ evolution	5	[123]
Anatase and rutile	NaBH_4_	✔	Rhodamine B degradation	4	[127]
Anatase	H_2_	✔	Diclofenac degradation	5	[128]

Notes: N/S: not studied; ✔: yes.

## Data Availability

Not applicable.

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
