# Peer review of "Recent Advances in Black TiO2 Nanomaterials for Solar Energy Conversion"

_nanomaterials, 2023, doi:10.3390/nano13030468_

Round 1

Reviewer 1 Report

Referee report on manuscriptRecent Advances in Black TiO2 Nanomaterials for Solar Energy Conversion

It is not clear whether the authors consider this manuscript as a review or a regular article.

Nevertheless, this is an interesting topic, which, of course, is needed in the development and promotion, the results that are discussed are interesting and can be accepted for publication after a more detailed disclosure of some ambiguities and uncertainties.

1. Introduction.  The information on TiO2 given here is clearly quite detailed. It is known that titanium oxide has several crystalline forms, and its functional properties also depend on its preparation, dimensions, and morphology. However, note here that there is a lot of very new research on TiO2, published by MDPI, see for example:

Serga, V.; Burve, R.; Krumina, A et al . Extraction–Pyrolytic Method for TiO2 Polymorphs Production. Crystals 202111, 431. https://doi.org/10.3390/cryst11040431

Trestsova, M.A.; et al . Oxidative C-H/C-H Coupling of Dipyrromethanes with Azines by TiO2-Based Photocatalytic System. Synthesis of New BODIPY Dyes and Their Photophysical and Electrochemical Properties. Molecules 202126, 5549. https://doi.org/10.3390/molecules26185549

Tsebriienko, T.; Popov, A.I. Effect of Poly(Titanium Oxide) on the Viscoelastic and Thermophysical Properties of Interpenetrating Polymer Networks. Crystals 202111, 794. https://doi.org/10.3390/cryst11070794

2.  More information can be found at https://www.mdpi.com/search?q=TiO2+and+nano+

3.  Fig.3 needs improvement because small details are not visible.

4.  the results discussed do not contain any information about aging, long-term stability, and how much the results are repeated from measurement to measurement in the case of measuring photocatalytic activity.

5. About porosity. Please discuss in what structure the pores appear more efficiently and what is the effect of the surface here. Are the pores located closer to the surface or so it is impossible to assert?

6. In the case of analysis of the optical properties, what is the actual concentration of point defects (oxygen vacancies), and how they were taken into account in the analysis of band gap parameters?  Is the decrease in Eg related to the defectiveness of the samples or the formation of real new structures?

7. the big drawback of this review is the lack of summary tables, which always allow you to see trends better than just text with references. 

Author Response

Reviewer 1#

It is not clear whether the authors consider this manuscript as a review or a regular article. Nevertheless, this is an interesting topic, which, of course, is needed in the development and promotion, the results that are discussed are interesting and can be accepted for publication after a more detailed disclosure of some ambiguities and uncertainties.

  1. Introduction.  The information on TiO2 given here is clearly quite detailed. It is known that titanium oxide has several crystalline forms, and its functional properties also depend on its preparation, dimensions, and morphology. However, note here that there is a lot of very new research on TiO2, published by MDPI, see for example:

Serga, V.; Burve, R.; Krumina, A et al. Extraction–Pyrolytic Method for TiO2 Polymorphs Production. Crystals 2021, 11, 431. https://doi.org/10.3390/cryst11040431

Trestsova, M.A.; et al. Oxidative C-H/C-H Coupling of Dipyrromethanes with Azines by TiO2-Based Photocatalytic System. Synthesis of New BODIPY Dyes and Their Photophysical and Electrochemical Properties. Molecules 2021, 26, 5549. https://doi.org/10.3390/molecules26185549

Tsebriienko, T.; Popov, A.I. Effect of Poly(Titanium Oxide) on the Viscoelastic and Thermophysical Properties of Interpenetrating Polymer Networks. Crystals 2021, 11, 794. https://doi.org/10.3390/cryst11070794

Response: We thank the referee’s for this valuable comment. More new research about TiO2 have been added as follows: Three main kinds of TiO2 including anatase, rutile, and brookite can be distinguished according to their different crystal structures [11,12,13]. Anatase and rutile are the most frequently investigated TiO2 photocatalysts because of their superior photocatalytic activity under UV irradiation than brookite. Photocatalytic performance and properties of TiO2 are severely influenced by its preparation, morphology, and dimensions. Serga et al. reported an extraction-pyrolytic method for the synthesis of nanocrystalline TiO2 powders using valeric acid as an extractant [14]. This method can be applied for the fabrication of anatase, rutile, or mixed anatase-rutile TiO2 powders [14]. Poly (titanium dioxide) is found to have significant influence on component compatibility and relaxation behavior of interpenetrating polymer networks [15]. TiO2 photocatalysts treated at 800 °C in hydrogen atmosphere for 1 h showed higher visible photocatalytic activity for C-H/C-H coupling of dipyrromethanes with azines than commercial TiO2 (P25) [16]. TiO2 nanosheet were proved to exhibit superior photocatalytic activity for CO2 reduction than nanoparticle thanks to its much higher surface area and surface activity [17]. In addition, the effect of particle size of TiO2 on photocatalytic pollutants removal were thoroughly investigated by Kim et al. [18]. Photocatalytic degradation efficiency for methylene blue can be effectively improved by controlling the particle size and TiO2 concentration in the reaction mixture [18]. These contents had been added on page 3 and page 4 in the revised manuscript.

  1. More information can be found at https://www.mdpi.com/search?q=TiO2+and+nano+

Response: We thank the referee for this nice information. More information about TiO2 together with the first suggestion has been added on page 3 and page 4 in the revised manuscript.

  1. Fig.3 needs improvement because small details are not visible.

Response: We thank the referee for this suggestion. Fig. 3 has been improved on page 7 in the revised manuscript.

  1.  The results discussed do not contain any information about aging, long-term stability, and how much the results are repeated from measurement to measurement in the case of measuring photocatalytic activity.

Response: We thank the referee for this comment. The aging, long-term stability, and the experimental conditions were discussed and summarized in the revised manuscript. The aging of the black TiO2 materials was generally not mentioned in the reported publications. The photocatalytic stability of hydrogen generation was mostly measured in 5 cycling tests within 15 h using the same solution. Some photocatalysts were tested in 10 repeated photocatalytic hydrogen formation measurements for more than 20 h. Black TiO2 photocatalysts usually showed good stability and light corrosion resistance in photocatalytic H2 evolution reaction, providing the possibility of long-term usage of black TiO2 photocatalysts for H2 production. The long-term photocatalytic stability of pollutants removal is often overlooked and unverified in most reported researches. Therefore, researchers should pay more attention to the aging, and photocatalytic stability in pollutants degradation in the future. The concluded information about the aging, stability, and repeated condition were added on page 29, 30, 31, 32, and 33 in the revised manuscript.

  1. About porosity. Please discuss in what structure the pores appear more efficiently and what is the effect of the surface here. Are the pores located closer to the surface or so it is impossible to assert?

Response: We thank the referee for this constructive comment. Porous black TiO2 photocatalysts tended to appear in three dimensional structures such as foams, pillars, and hollow structures. Zhang et al. synthesized 3D macro-mesoporous black TiO2 foams via freeze-drying, cast molding technology, and high-temperature surface hydrogenation [65]. Large closed pores were generated using polyacrylamide as organic template while plenty of open pores were formed in frameworks and on surface of black TiO2 thanks to the water evaporation in freeze-drying process [65]. This black TiO2 material exhibited self-floating amphiphilic property, and enhanced solar energy harvesting efficiency [65]. Zhou et al. prepared porous black TiO2 pillars through oil bath reaction and high-temperature hydrogenation reduction [66]. The porous structure and mesopores of black TiO2 pillars were clearly observed by Scanning electron microscope and transmission electron microscope [66]. The enhanced photocatalytic performance was attributed to more active surface sites offered by the porous pillar structure and the self-doped Ti3+ [66]. Hollow structured black TiO2 with plenty of pore channels and exposed surface also showed enhanced photocatalytic efficiency [57]. The pores of porous black TiO2 generally located in its whole frameworks with open pores connected with surface which provide abundant of active sites and surface defects, thus promoting photocatalytic performance. Ethylenediamine was often utilized to maintain the porous structure of black TiO2 and to prevent its phase transformation from anatase to rutile. This has been added on page 8 and page 9 in the revised manuscript.

  1. In the case of analysis of the optical properties, what is the actual concentration of point defects (oxygen vacancies), and how they were taken into account in the analysis of band gap parameters?  Is the decrease in Eg related to the defectiveness of the samples or the formation of real new structures?

Response: We thank the referee for this valuable comment. Black TiO2 materials generally possessed certain amounts of oxygen vacancies which can be confirmed by X-ray photoelectron spectroscopy (XPS). The concentration of oxygen vacancies was normally controlled by the different thermal treatment time or temperature [41,45,51]. However, there is lack of precise, quantitative characterization techniques for oxygen vacancies present on black TiO2 surface. The band gap parameters of black TiO2 were usually measured and calculated by XPS and UV-vis spectroscopy measurements. The decrease in Eg of black TiO2 was assumed to be related to surface disorder including the presence of Ti3+ and oxygen vacancies [47]. This has been added on page 6 and page 7 in the revised manuscript.

  1. The big drawback of this review is the lack of summary tables, which always allow you to see trends better than just text with references. 

Response: We thank the referee for this valuable suggestion. Table 1 and Table 2 have been added on page 9 and 33, respectively in the revised manuscript as follows:

Table 1 Properties of black TiO2 nanomaterials.

Crystal phase

Reducing agent

Color

Morphology

Oxygen vacancies

Ref.

Anatase with minor rutile

NaBH4

Black

Nanospheres

✔

26

Anatase and rutile

 hydrogen radicals

Black

Nanospheres

✔

27

Anatase

NaBH4

Black

Nanoparticles

✔

28

Anatase and rutile

hydrogen

Black

microspheres

✔

29

Anatase and rutile

 Pulsed

Laser Ablation in Liquid

Black

Core-shell microspheres

✔

30

Anatase

 Urea

Black

Nanoparticles

✔

31

Anatase

NaBH4

Black

Nanospheres

✔

32

Anatase

NaBH4

Black

Nanospheres

✔

33

Anatase

NaBH4

Black

Nanospheres

✔

34

Anatase

Electrochemical reduction

Black

Nanotubes

✔

35

Anatase

CaH2

Black

Nanotubes

✖

36

Anatase

hydrogen

Black

Nanotubes

✔

37

Anatase

Ag particles

Black

Nanotubes

✔

38

Anatase with minor rutile

Electrochemical reduction

Black

Nanotubes

✔

39

Anatase and rutile

Electrochemical reduction

Blue and Black

Nanotube arrays

N/S

40

Anatase and rutile

Electrochemical reduction

Dark yellow

Nanotube arrays

✔

41

Anatase and rutile

aluminothermic reduction

Black

Nanotubes

✔

42

Anatase

Electrochemical reduction

Black

Nanotube arrays

✔

43

Anatase

Hydrogen

Black

Nanowires

✔

44

Anatase

NaBH4

Black

Nanowires

✔

45

Anatase

Glycerol

Black

Nanoparticles and nanowires

✔

46

Rutile

NaBH4

N/S

Hollow nanowires

✔

47

Anatase

H2S and SO2

Black

Nanoplatelets

✔

48

Anatase

NaBH4

Black

Nanoplates

✔

49

Anatase

Hydrogen

Black

Nanosheets

✔

50

Anatase

Aluminothermic reduction

Black

Nanosheet array films

✔

51

Anatase

Hydrogen

Black

Nanobelts

✔

52

Anatase

NaBH4

Black

Nanobelts

✔

53

Anatase

Hydrogen

Black

Nanocages

✔

54

Anatase

Hydrogen

Black

3D nanoflower

✔

55

Anatase and rutile

NaBH4

Black

Nanofiber

✔

56

Anatase

NaBH4

Black

Hollow shell

✔

57

Anatase

Annealing in vacuum

Black

Core-shell

✔

58

Anatase

Hydrogen

Black

Mesoporus hollow sphere

✔

59

Anatase and rutile

Electrochemical

Black

Films

✔

60

Anatase

H2 plasma treatment

Black

Mesoporous films

✔

61

Anatase

Hydrogen

Black

Hierarchical nanolace films

✔

62

Notes: N/S: not studied; ✔: yes; ✖: no.

Table 2 Applications and properties of black TiO2 nanomaterials.

Black TiO2

Reducing agent

Oxygen vacancy

Application

Numbers of cycling tests

Ref.

Anatase with minor rutile

NaBH4

Tetracycline degradation

5

26

Anatase and rutile

H2

Rhodamine B degradation

N/S

77

Anatase

H2

H2 evolution

10

79

Anatase

H2

H2 evolution

5

80

Anatase

H2

H2 evolution

5

81

Anatase

H2

H2 evolution

5

83

Anatase

H2

Tetracycline degradation

4

84

Anatase

H2

N/S

Orange G degradation

N/S

85

Rutile

NaBH4

H2 evolution

4

87

Anatase with minor rutile

H2

H2 evolution

10

89

Anatase

Anodization

Rhodamine B degradation

N/S

90

Anatase

H2

N/S

H2 evolution

5

127

Anatase and rutile

NaBH4

Rhodamine B degradation

4

131

Anatase

H2

Diclofenac degradation

5

132

Notes: N/S: not studied; ✔: yes; ✖: no.

Reviewer 2 Report

Dear Editor: I would like to express my deep thanks for inviting me to review the manuscript ID: nanomaterials-2160948-peer-review-v1

Title:      Recent Advances in Black TiO2 Nanomaterials for Solar Energy Conversion

Authors:  Lijun Liao, Mingtao Wang, Zhenzi Li, Xuepeng Wang, Wei Zhou

Comments:

Overall, this review article is well organized and explained in detail. However, some minor corrections are necessary before accepting this manuscript.

1)        In abstract section, it is necessary to explain the objects of this reviews.

2)        The synthesis process of Black TiO2 Nanomaterials is not sufficient. Need to describe in details with other methods.

3)        Figure 1 does not provide any information, so please delete it.

4)        Photocatalytic degradation of pollutants is not sufficient. Please explain in detail? Follow below references and cited them

1.         M. Zeshan, I. A. Bhatti, M. Mohsin, M. Iqbal, N. Amjed, N. AlMasoud,T.S. AlomaRemediation of pesticides using TiO2 based photocatalytic strategies: A review” Chemosphere 300 (2022) 134525

2.         B.T. Lee, J.K. Han, A.K. Gain, K.H. Lee, F. Saito, “TEM microstructure characterization of nano TiO2 coated on nano ZrO2 powders and their photocatalytic activity” Materials Letters 60 (17-18), (2006) 2101-2104

3.         M.A.E. Wafi, M.A. Ahmed, H. S. Abdel-Samad, H.A.A Medien, “Exceptional removal of methylene blue and p-aminophenol dye over novel TiO2/RGO nanocomposites by tandem adsorption-photocatalytic processes” Materials Science for Energy Technologies 5 (2022) 217-231

Summary and outlook part:

Please concise the Summary and outlook part.

RECOMMENDATION

After reviewing the enclosed manuscript for “Nanomaterials”, the present manuscript contains some kinds of scientific analysis but it is mandatory required to modify according to the preceding remarks. So, the manuscript can be publication after major revision.

Author Response

Reviewer 2#

Overall, this review article is well organized and explained in detail. However, some minor corrections are necessary before accepting this manuscript.

1)        In abstract section, it is necessary to explain the objects of this reviews.

Response: We thank the referee for this valuable suggestion.  This review represents an attempt to conclude the recent developments in black TiO2 nanomaterials synthesized by modified treatment which presented different structure, morphological features, reduced band gap, and enhanced solar energy harvesting. Special emphasis has been given to the newly developed synthetic methods, porous black TiO2, and the approaches for further improving photocatalytic activity of black TiO2. Various black TiO2, doped black TiO2, metal-loaded black TiO2 and black TiO2 heterojunction photocatalysts, and their photocatalytic applications and mechanisms in the field of energy and environment are summarized in this review, to provide useful insights and new ideas in the related field. This has been added on page 1 and page 2 in the revised manuscript.

2)        The synthesis process of Black TiO2 Nanomaterials is not sufficient. Need to describe in details with other methods.

Response: We thank the referee for this suggestion. More information about the synthesis of black TiO2 nanomaterials are concluded as follows:

3.3 Hot-wire annealing method

In addition to high temperature hydrogenation reduction and solid-phase reduction, researchers have also explored some other methods to synthesize black TiO2, which has made the method of preparing black TiO2 diversified. Wang et al. proposed a simple and direct hot-wire annealing (HWA) method [72]. The titanium dioxide nanorods were treated with highly active atomic hydrogen simply generated by hot wire [72]. The reduction mechanism was similar to that of high-temperature hydrogenation [72]. The resulted black TiO2 nanorods had better stability and higher photocurrent density compared with the traditional hydrogenation method [72]. In addition, it had no damage to the photoelectric chemical devices [72].

3.4 Anode oxidation method

  The introduction of crystal defects to titanium dioxide can effectively extend light absorption range to the visible light region without side effects. Anode oxidation is a simple and efficient method to synthesize defective black TiO2. Dong et al. successfully prepared black TiO2 using a two-step anode oxidation method [88]. The first step was to anodize Ti foil in ethylene glycol solution with a certain proportion of NH4F and distilled water, and the corresponding voltage was set at 60 V [88]. After 10 hours of oxidation, an oxide layer was obtained [88]. Subsequently, the Ti foil was purified to remove organic impurities, and treated at high temperature (450 ℃) for 1 h to form black TiO2 [88].

3.5 Plasma treatment

  Zhu et al. prepared black TiO2 nanoparticles via one-step solution plasma method under mild conditions [27]. Structural disorder layer was assumed to be formed in TiO2 after solution plasma process [27]. The light absorption of TiO2 in visible and near infrared range was significantly enhanced after the plasma treatment, thus increasing its activity in water evaporation under solar illumination [27]. Teng et al. prepared black TiO2 using P25 as the precursor system, hydrogen plasma, and a hot filament chemical vapor deposition (HFCVD) device with H2 as the reducing gas [75]. Visible and near infrared light absorption of TiO2 were much enhanced after surface reduction [75]. Oxygen vacancies and Ti-H bonds were formed on black TiO2 surface, thereby improving photocatalytic activity [75].

3.6 Gel combustion

Ullattil et al. prepared black anatase TiO2-x photocatalysts through a one pot gel combustion process using titanium butoxide, diethylene glycol, and water as precursors [89]. Plenty of Ti3+ and oxygen vacancies existed in the synthesized black anatase TiO2 nanocrystals confirmed by XPS measurements [89]. The light absorption of TiO2 was extended from UV to near infrared range [89]. Campbell et al. also synthesized black TiO2 via sol-gel combustion method using titanium tetraisopropoxide as precursor [90]. The light absorption ability was significantly enhanced compared to commercial TiO2 [90]. The obtained black TiO2 with high surface area demonstrated much improved photocatalytic degradation efficiency of organic dye under visible light irradiation [90]. This has been added on page 16 and page 17 in the revised manuscript.

3)        Figure 1 does not provide any information, so please delete it.

Response: We thank the referee for this comment. Figure 1 has been deleted in the revised manuscript.

4)        Photocatalytic degradation of pollutants is not sufficient. Please explain in detail? Follow below references and cited them

  1. M. Zeshan, I. A. Bhatti, M. Mohsin, M. Iqbal, N. Amjed, N. AlMasoud,T.S. Aloma “Remediation of pesticides using TiO2 based photocatalytic strategies: A review” Chemosphere 300 (2022) 134525
  2. B.T. Lee, J.K. Han, A.K. Gain, K.H. Lee, F. Saito, “TEM microstructure characterization of nano TiO2 coated on nano ZrO2 powders and their photocatalytic activity” Materials Letters 60 (17-18), (2006) 2101-2104
  3. M.A.E. Wafi, M.A. Ahmed, H. S. Abdel-Samad, H.A.A Medien, “” Materials Science for Energy Technologies 5 (2022) 217-231

Response: We thank the referee for this constructive comment. The part of photocatalytic degradation of pollutants has been improved as follows: In addition to hydrogen production, pollutants degradation is also one of the main applications of photocatalysis. Titanium dioxide was often used in the photocatalytic degradation of organic dye and pesticides [124,125,126]. Black TiO2 with enhanced light absorption ability would have much improved photocatalytic pollutants removal efficiency. Black TiO2 obtained by CaH2 reduction not only presented enhanced hydrogen generation rate which was 1.7 times that of the original sample, but also achieved a huge improvement in the degradation of pollutants with completely removal of methyl orange within 8 min [71]. Hamad et al. synthesized black TiO2 by using a new method of controlled hydrolysis [83]. The oxygen vacancy concentration was significantly increased with much reduced band gap, thereby showing excellent organic pollutants degradation rate under visible light irradiation [83].

Oxygen vacancy plays an important role in photocatalysis. Black TiO2 prepared by Teng et al. via vapor deposition had high photocatalytic oxidation activity for organic pollutants in water due to the formation of Ti-H bonds and a large number of oxygen vacancies [75]. All pollutants (rhodamine B) could be completely degraded within 50 min detected by the UV-vis spectrophotometer [75]. The defective TiO2-x prepared by anodic oxidation method was characterized by electron paramagnetic resonance spectroscopy, confirming the existence of oxygen vacancies and extension of the absorption from ultraviolet to visible light region [88]. This black TiO2 material showed excellent photocatalytic degradation activity for rhodamine B under 400-500 nm light irradiation [88].

Black TiO2 based heterojunction could significantly improve its photocatalytic efficiency in pollutants remediation thanks to the enhanced charge separation and transfer efficiency. Jiang et al. prepared black TiO2/Cu2O/Cu composites via in-situ photodeposition and solid reduction method [127]. The light energy harvesting in visible and infrared range was much enhanced after the formation the composites [127]. The photocatalytic efficiency of the composites for Rhodamine B degradation was improved compared to commercial P25 due to the enhanced charge separation efficiency [127]. Qiang et al. synthesized RuTe2/black TiO2 photocatalyst through gel calcination and microwave-assisted process [128]. The light absorption range of the as-made composites was enlarged compared to the pristine TiO2. The photocatalytic efficiency of diclofenac degradation was 1.2 times higher than pure black TiO2 [128]. The stability of RuTe2/black TiO2 for photocatalytic diclofenac degradation was confirmed via 5 repeated experiments [128].

   Tetracycline is a toxic antibiotic which is difficult to remove. Li et al synthesized black TiO2 modified with Ag/La presented improved visible light photocatalytic performance for tetracycline degradation [26]. The photocatalytic stability and reusability of black TiO2 based photocatalysts were studied via 5 cycling tests without apparent deactivation [26]. Wu et al. reported that the synthesized black anatase TiO2 exhibited impressive photocatalytic degradation of tetracycline [82]. Its degradation efficiency of tetracycline was 66.2 % under visible light illumination, which was higher than that of white titanium dioxide and doped titanium dioxide [82]. In addition, ·O2− and h+ were found to play important roles in the degradation process, which was different from original TiO2, providing new insight for environmental protection [82]. The stability of photocatalytic tetracycline degradation was measured in four repeated experiments within 960 min without apparent deactivation after four cycles [82]. The long-term photocatalytic stability of pollutants removal is often overlooked and unverified in most reported researches. Therefore, researchers should pay more attention to the aging, and photocatalytic stability in pollutants degradation in the future. And the above references have been cited in the revised manuscript. This has been added on page 31, page 32, page 33, page 34 in the revised manuscript.

Summary and outlook part:

Please concise the Summary and outlook part.

Response: We thank the referee for this comment. The Summary and outlook part has been improved on page 34 in the revised manuscript.

Reviewer 3 Report

Review of the manuscript titled: “Recent Advances in Black TiO2 Nanomaterials for Solar Energy Conversion” written by Lijun Liao, Mingtao Wang, Zhenzi Li, Xuepeng Wang, Wei Zhou

The authors have made a great effort to gather recent literature on the black TiO2. The article can be published after some minor revisions.

The authors must polish the text and correct some minor errors. For example, in the sentence: “has been developed and attracted much attentions” the plural of attentions is not needed. The sentence “Various morphology of black TiO2…” must be rephrased. The sentence “or other reducibility materials” must be rephrased. The authors should correct for grammar mistakes and rephrase some sentences.

The authors should add and discuss more recent articles from 2023 and 2022 such as: https://onlinelibrary.wiley.com/doi/abs/10.1002/solr.202200929

https://www.sciencedirect.com/science/article/pii/S0169433222027581

https://pubs.rsc.org/en/content/articlehtml/2022/cy/d2cy01543a

The authors should search the literature for more recent articles.

Finally, the authors should add the following relevant publications:

https://www.sciencedirect.com/science/article/pii/S016943321400590X

https://www.sciencedirect.com/science/article/pii/S0926337316300674

Author Response

Reviewer 3#

The authors have made a great effort to gather recent literature on the black TiO2. The article can be published after some minor revisions. The authors must polish the text and correct some minor errors. For example, in the sentence: “has been developed and attracted much attentions” the plural of attentions is not needed. The sentence “Various morphology of black TiO2…” must be rephrased. The sentence “or other reducibility materials” must be rephrased. The authors should correct for grammar mistakes and rephrase some sentences.

Response: We thank the referee for this valuable comment. The sentence “has been developed and attracted much attentions” has been corrected into “has been developed and attracted much attention” on page 3 in the revised manuscript. The sentence “Various morphology of black TiO2…” has been changed into “black TiO2 with different morphology…” on page 5 in the revised manuscript. The sentence ‘Materials used in solid phase reduction method are generally NaBH4 [71], CaH2 [72], Mg powder [72], or other reducibility materials [72].’ has been changed into “Different reducing agents have been utilized in solid phase reduction method including NaBH4 [71], CaH2 [72], Mg powder [72], and other compounds [72].” on page 9 in the revised manuscript. The whole manuscript has been checked for the correction of grammar mistakes in the revised manuscript.

The authors should add and discuss more recent articles from 2023 and 2022 such as: https://onlinelibrary.wiley.com/doi/abs/10.1002/solr.202200929

https://www.sciencedirect.com/science/article/pii/S0169433222027581

https://pubs.rsc.org/en/content/articlehtml/2022/cy/d2cy01543a

The authors should search the literature for more recent articles.

Finally, the authors should add the following relevant publications:

https://www.sciencedirect.com/science/article/pii/S016943321400590X

https://www.sciencedirect.com/science/article/pii/S0926337316300674

Response: We thank the referee for this comment. More recent articles including the above mentioned publications have been discussed and cited on page 3, page 4, page 11, page 17, page 20, page 24, and page 32 in the revised manuscript.

Round 2

Reviewer 1 Report

The authors have strongly improved their original manuscript which now can be recommended for publication.

Reviewer 2 Report

Authors addressed all the review comments in the revised manuscript.